# Cavity-mediated exciton hopping in a dielectrically engineered polariton system

Lukas Husel [1] ✉, Farsane Tabataba-Vakili [1,2,3], Johannes Scherzer [1],
Lukas Krelle[1,7], Ismail Bilgin[1], Samarth Vadia[1], Kenji Watanabe [4],
Takashi Taniguchi [5], Iacopo Carusotto [6] & Alexander Högele [1,2] ✉

Exciton-polaritons – coherently hybridized states of excitons and photons – are instrumental for solid-state nonlinear optics and quantum simulations. To enable engineered polariton energy landscapes and interactions, local control over the particle-like states can be achieved by tuning the properties of the exciton constituent. Monolayer transition metal dichalcogenides stand out in this respect, as they readily allow for a deterministic, flexible and scalable control of excitons, and thus of hybrid exciton-polaritons, via environmental dielectric engineering. Here, we demonstrate the realization of mesoscopic exciton-polariton domains in a structured dielectric exciton environment, and establish an effective long-range exciton hopping in the dispersive regime of cavity-coupling. Our results represent a crucial step toward interacting polaritonic networks and quantum simulations in exciton-polariton lattices based on dielectrically tailored two-dimensional semiconductors.

Coherently hybridized states of cavity photons and semiconductor excitons manifest as exciton-polaritons and underpin phenomena of non-equilibrium quantum many-body physics[1,2] and quantum simulations[3] in conventional quantum well microcavity structures. In two-dimensional semiconductors and related van der Waals hetero-structures, large exciton binding energies and oscillator strengths enable robust polariton formation and condensation at elevated temperatures[4,5] with unique features provided by spin-valley locking[6,7], novel quasiparticles with enhanced nonlinearities upon doping[8], or moiré exciton-polaritons in heterostructures[9,10]. These properties establish semiconductor monolayers and heterostructures as prime candidates for studies of strong light-matter coupling in the solid-state.

Control over the polariton degrees of freedom via the excitonic fraction is the key to recent developments in device design[11] and has enabled demonstrations of topological insulators[12] or Kardar-Parisi-Zhang universality[13] in conventional semiconductor quantum wells.

Within this framework, excitons in monolayer transition metal dichalcogenides (TMDs) with high sensitivity to their dielectric environment provide a means of additional engineering, taking advantage of sizable changes in the band gap and binding energy induced by proximal dielectrics[14–17]. As such, local disorder in TMD devices facilitates trapping of polaritons with enhanced coherence[5,18], while complementary approaches have demonstrated polariton trapping by additional monolayers[19,20], laser-induced disorder[21], and local strain[22]. Despite this progress, deterministic, scalable, and tunable control over local exciton-polariton energies, potentials, and couplings remains challenging.

Here, we demonstrate control of local exciton-polariton energies via environmental dielectric engineering and spectral cavity detuning, and show how the spatially confined cavity mode can be used to mediate hopping-like polariton coupling. To define local exciton sites, we use a nanopatterned encapsulation layer of hexagonal boron nitride (hBN), which consolidates disk-shaped exciton domains with

[1]Fakultät für Physik, Munich Quantum Center, and Center for NanoScience (CeNS), Ludwig-Maximilians-Universität München, München, Germany. [2]Munich Center for Quantum Science and Technology (MCQST), München, Germany. [3]Institute of Condensed Matter Physics, Technische Universität Braunschweig, Braunschweig, Germany. [4]Research Center for Electronic and Optical Materials, National Institute for Materials Science, Tsukuba, Japan. [5]Research Center for Materials Nanoarchitectonics, National Institute for Materials Science, Tsukuba, Japan. [6]Pitaevskii BEC Center, INO-CNR and Dipartimento di Fisica, Universita di Trento, Trento, Italy. [7]Present address: Institute for Condensed Matter Physics, TU Darmstadt, Darmstadt, Germany. ✉e-mail: lukas.husel@physik.lmu.de; alexander.hoegele@lmu.de

energies distinct from excitons in the surrounding monolayer regions with unpatterned hBN. Strong coupling of one or multiple exciton domains to the optical mode of a fiber-based microcavity results in distinct polariton states, with properties determined by both the locally engineered dielectric environment and the actively controlled spectral cavity-coupling. The resulting modulation in the lower polariton energy landscape is evidenced in transmission spectroscopy of the strongly-coupled fiber cavity. Moreover, we demonstrate how the regime of dispersive cavity coupling[23] mediates effective hopping between distant domains of excitons weakly dressed by cavity photons as a premise to quantum simulation architectures, complementing those already established for superconducting qubits[24], ultracold atoms[25], solid-state quantum emitters[26], and optomechanical systems[27].

## RESULTS

Our approach, illustrated schematically in Fig. 1a, is based on a van der Waals heterostack with a monolayer of molybdenum diselenide (MoSe$_2$) encapsulated by a planar bottom hBN layer and a patterned top hBN layer with through-holes defined by reactive-ion etching (see the "Methods" section for details). The resulting modification of the dielectric environment on the scale of a few hundred of nanometers affects both the monolayer band gap and the exciton binding energy[14,15,17], yielding a spatially modified exciton resonance energy which in turn determines local properties of exciton-polaritons upon strong coupling to an optical microcavity with photonic modes confined in all spatial directions. The left panel in Fig. 1b shows an optical micrograph of the corresponding sample, with two holes in the top hBN layer of same diameter and variable distance as pairs of yellow circles. The right panel illustrates schematically the resulting

exciton landscape: left ($L$) and right ($R$) disk-shaped exciton domains at each etch site are surrounded by monolayer excitons ($X$) in the unpatterned area. Due to fabrication imperfections such as interfacial bubbles and unintentional strain, the exciton energies $E_L$ and $E_R$ differ between the two nominally identical domains, and both differ from the surrounding exciton energy $E_X$ by virtue of different dielectric environments.

In the following, we focus on three etch-site pairs $P_1$, $P_2$ and $P_3$ in Fig. 1b with 2.0, 1.4 and 1.1 μm distances between the centers of the left and right holes with identical diameters of 0.6 μm. The finite extent of the fundamental Gaussian cavity mode with a waist of ~1μm (shown by the red dashed circle in the schematics of Fig. 1b) and the tunability of the resonance energy as well as the lateral mode position of our open cavity (as indicated in Fig. 1c) allow us to study different limits of polaritons in strong light-matter coupling. First, by placing the cavity mode over the left site of the pair $P_1$ or $P_2$ that is sufficiently distant from its right counterpart, we study the local formation of polaritons in the left disk as well as their coupling to the surrounding exciton-polariton continuum (top panel of the schematics in Fig. 1b). In the second setting (central panel in Fig. 1b), the cavity mode creates and samples both left and right polariton disks, yet at a distance too large for intersite coupling. The third configuration (bottom panel in Fig. 1b), finally, is used to demonstrate effective cavity-mediated coupling between the left and right sites of polariton pairs.

We begin by calibrating the coupling strength between the exciton domains and our tunable microcavity in a closed-cycle cryostat with a base temperature of 4.3 K[28] according to the schematics in Fig. 1c (see the "Methods" section for details on the cavity setup). The nanopatterned heterostack is placed on a macroscopic planar mirror, whose vertical separation $L_C$ from the microscopic mirror of the

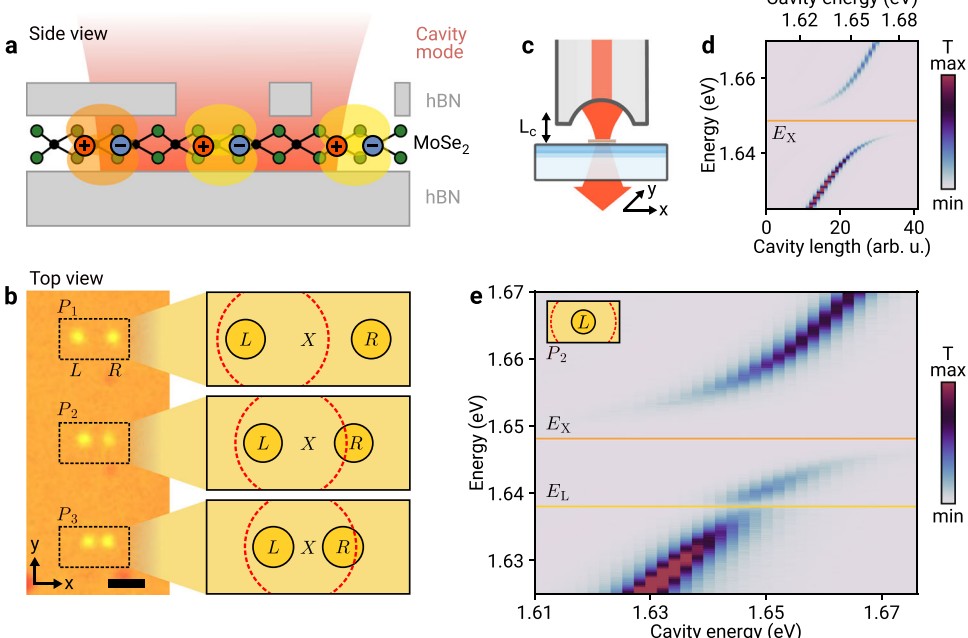

**Fig. 1 | Dielectric engineering of polariton domains. a** Schematics of a MoSe$_2$ monolayer encapsulated by planar bottom and patterned top hBN layers with spatially distinct regions of exciton-polaritons formed by strong coupling to the mode of an open cavity. **b** Left panel: optical micrograph of the van der Waals heterostructure, with pairs of holes in the top hBN layer $P_1$, $P_2$, and $P_3$ visible as yellow circles of nominally identical left ($L$) and right ($R$) disks with variable separation, surrounded by fully encapsulated monolayer. The scale bar is 2 μm. Right panel: The red dashed circle shows the waist diameter of the cavity mode on the scale of the diameters and distances of disk-shaped domains surrounded by regions of monolayer excitons ($X$) in unpatterned hBN. **c** Schematic of the

cryogenic fiber-based open microcavity, with piezoelectrically actuated lateral translation in the $x-y$ plane and cavity length ($L_C$) tuning along $z$. **d** Cavity transmission as a function of the cavity length tuned via the $z$-piezo voltage, recorded away from structured hBN. The cavity energy corresponding to each length index is indicated on the upper horizontal axis. The exciton energy $E_X$ is indicated by the horizontal solid line. **e** Same but with the cavity mode positioned near the center of a single exciton disk of etched pair $P_2$ as illustrated in the inset. The solid horizontal lines indicate the energy of monolayer and disk-localized excitons $E_X$ and $E_L$, respectively.

micromachined fiber facet is controlled by piezoelectric actuators, which also allow for lateral displacement of the sample with respect to the cavity mode. As such, the spectral resonance condition between the exciton and cavity energy, $E_X$ and $E_C$, is tunable via the cavity length and exhibits a clear signature of strong-coupling in the cavity transmission of Fig. 1d on a region away from structured hBN. The avoided crossing is a hallmark of polariton formation, with light-matter coupling strength $g_X = 9.6$ meV at longitudinal cavity mode order $q = 6$ as determined from the dissipative model analysis (see Supplementary Note II for details). This coupling strength is characteristic for cavity-coupling of monolayer excitons[28,29], and places the system together with polariton linewidths of ~2.5 meV and the cavity linewidth $\kappa \simeq$ 1.5 meV (limited by residual vibrational fluctuations in the cavity length[28]) in the regime of strong light-matter coupling[30].

We observe a strong modification of the characteristic exciton-polariton splitting as we position the cavity over the left hole-etched site of $P_2$, with cavity transmission shown in Fig. 1e. As a function of the cavity length detuning, we observe an additional polariton branch related to the disk-localized exciton fraction with energy $E_L$, redshifted from $E_X$ by 10 meV due to effectively reduced screening below the hole in hBN. As the area of the local exciton domain is smaller than the cavity spot, the corresponding light-matter coupling strength is reduced to $g_L = 2.65 \pm 0.04$ meV as compared to the coupling of spatially unconfined monolayer excitons in regions with both-sided hBN encapsulation (see Supplementary Note III for details). This scaling of the light-matter coupling with the spatial extent of exciton-confining domains with redshifted transition energy indicates the formation of local polariton disks.

The creation of exciton domains results in local shifts of the polariton energy, which we map out by spatial and spectral cavity tuning with data shown in Fig. 2. Displacing the cavity laterally at a constant cavity length across one domain of the pair $P_1$, the cavity transmission leads to the evolution of the upper and lower polariton branches as in Fig. 2a. In agreement with the data in Fig. 1e, the middle polariton branch is visible at an energy of ~1.640 eV. The intensity of this feature is brightened as a result of spectral overlap with higher order transverse cavity modes (see Supplementary Note II for details). Crucially, the lower polariton branch exhibits an energetic minimum at the center of the etch site. We emphasize that this redshift of ~2 meV would correspond to an attractive polariton potential for spatially extended polaritons in two-dimensional cavities[31], thereby providing lateral confinement for exciton-polaritons. The superimposed near-linear energy gradient, also evidenced in the upper polariton branch, stems from the unintentional spatial gradient of the exciton energy $E_X$ around this site.

The effect of the nanostructured dielectric environment on the polariton energy is even richer in the pair $P_2$ with transmission data in Fig. 2b. Upon spatial displacement across the left and right disk of the pair for a constant cavity energy, we observe in Fig. 2b two local energy minima for the lower polariton branch, indicated by the two Gaussian contributions (dashed lines) to the full transmission profile (solid line). The difference in the left and right local energy shift is related to differences in the disks from imperfect fabrication. Remarkably, the energy shift $\Delta E_{LP}$ is tunable via the cavity energy, as evident from the dependence of the lower polariton energies on the cavity energy shown for five discrete values in Fig. 2d. A systematic study of the lower

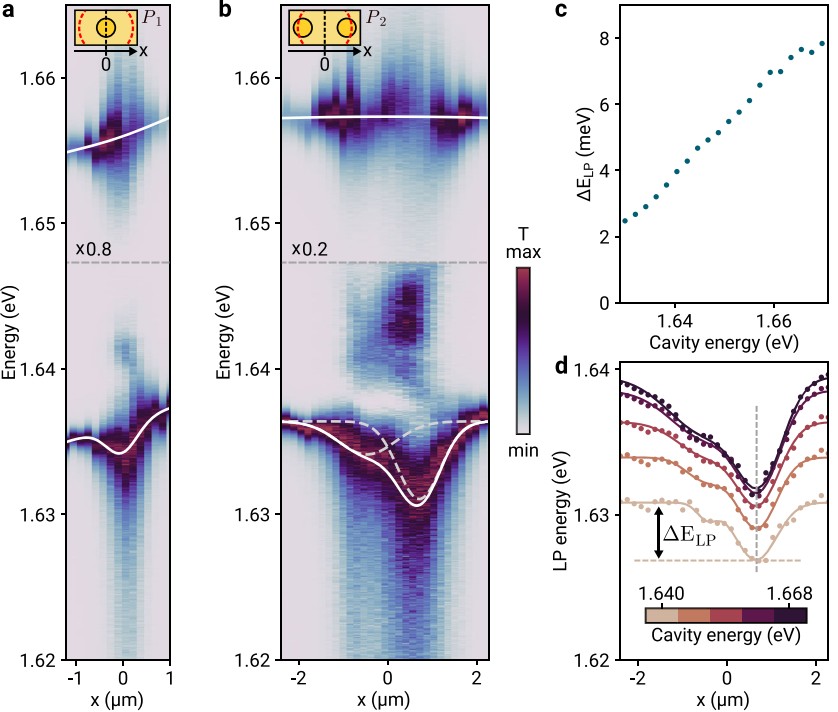

**Fig. 2 | Dielectrically engineered polariton disks with cavity-controlled local energy modulation. a** Cavity transmission for a fixed cavity energy and lateral cavity mode translation across the right exciton domain of pair $P_1$, as illustrated in the inset. The cavity energy corresponds to an exciton-cavity detuning $\Delta_X = -5$ meV at position $x = -1.2$ μm. The white solid lines show best fits of polynomial and Gaussian profiles to maximum transmission of the upper and lower polariton branches, respectively. **b** Same but for the domain pair $P_2$, recorded at similar detuning of $\Delta_X = -7$ meV at $x = -2.2$ μm ($x = 0$ μm corresponds to the cavity mode positioned in between the two exciton disks, as illustrated in the inset). The dashed lines are the individual Gaussian contributions to the full lower polariton transmission profile shown by the solid line. The spectra at each position were normalized to the maximum transmission of the lower polariton branch and rescaled for the upper polariton branch by a factor of 0.8 and 0.2 as indicated in the respective sub-panels. **c** Lower polariton energy shift $\Delta E_{LP}$ as a function of the cavity energy, defined as illustrated in (**d**) for the right polariton disk of $P_2$. **d** Energy profiles of the lower polaritons (corresponding to the transmission maxima in (**b**) as a function of the lateral cavity displacement for five cavity energies indicated by the color-bar. The solid lines show best fits of two Gaussians to the polariton energy profiles at different cavity energies.

polariton energy shift, shown in Fig. 2c, reveals a monotonous increase in $\Delta E_{LP}$ with increasing cavity energy, spanning a range of several meV.

The dependence of $\Delta E_{LP}$ on the cavity resonance condition can be understood by noting that for large cavity energy, the lowest polariton energy in the system is dictated by the lowest-energy exciton. By virtue of dielectric engineering, the exciton energy at the etch-site center is reduced from its monolayer value in fully hBN-encapsulated regions. Thus, in the limit of large cavity energies, the local energy shift for the lower polariton corresponds to the energy difference imprinted by different dielectric environments.

The second key feature of our system is the ability to establish site-to-surrounding and site-to-site coupling in the dispersive cavity regime, mediating an effective long-range hopping. From a theoretical perspective, this coupling follows from the time-independent Hamiltonian for multiple cavity-coupled exciton domains $i$ with light-matter coupling strengths $g_i$, energies $E_i$ and detunings $\Delta_i = E_i - E_C$ from the cavity energy $E_C$. Expansion to second order in $g_i/\Delta_i$ via a Schrieffer-Wolff transformation[23,24] yields:

$$H \simeq \left(E_C - \sum_i \frac{g_i^2}{\Delta_i}\right)a^\dagger a + \sum_i \left(E_i + \frac{g_i^2}{\Delta_i}\right)b_i^\dagger b_i + \sum_i \sum_{j \neq i} \frac{g_i g_j}{2\Delta_i}\left(b_i^\dagger b_j + b_j^\dagger b_i\right),$$

(1)

with the respective bosonic annihilation operators for cavity photons and excitons, $a$ and $b_i$. The result is a system described by multiple exciton resonances which are weakly dressed by cavity photons, evidenced by the diagonal first and second terms of Eq. (1). In addition, excitons associated with different resonances $i$ and $j$ are coupled via an effective beam-splitter type coupling of strength $J_{ij} = g_i g_j \left(\Delta_i^{-1} + \Delta_j^{-1}\right)/2$ as described by the third term of Eq. (1), mediated by dispersive exchange of cavity photons.

The resulting effective coupling between disk-localized excitons and their surrounding monolayer excitons is sizable when the cavity mode is positioned close to the edge of a single etch site. This effect is shown conceptually in Fig. 3a: since the light-matter coupling strength of $X$ excitons greatly exceeds that of $R$ and $L$ domains, the dispersive shift of magnitude $g_i^2/\Delta_i$ (see Eq. (1)) can be used to tune the difference in the respective energies of the bare excitons (left panel) into resonance (central panel). Here, dispersive coupling induces an effective splitting given by $2J_{LX}$ (right panel). The corresponding experiment on the left disk in pair $P_2$ shows in Fig. 3b the avoided crossing due to dispersive cavity-dressing and coupling, with a maximum coupling strength of $2J_{LX} = 1.5$ meV for optimal conditions. In this regime, the eigenstates of the system derive from the Hamiltonian of Eq. (1), with energies shown by the black dashed lines in the right panel of Fig. 3b

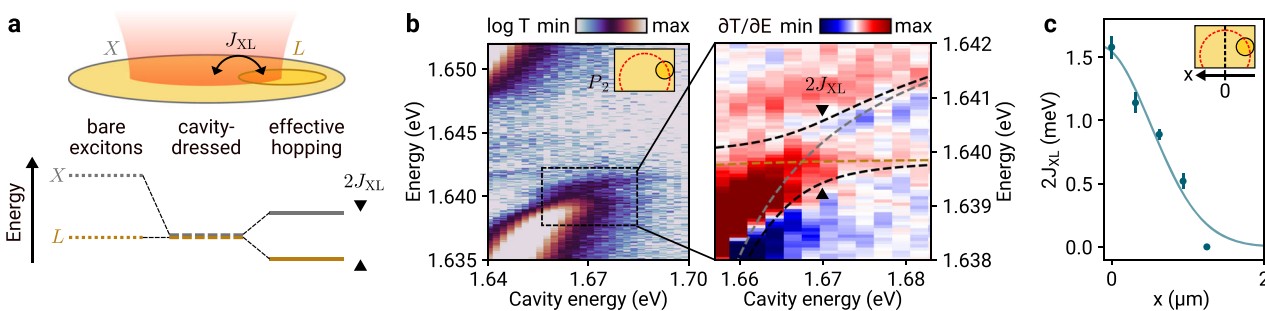

**Fig. 3 | Cavity-mediated long-range effective exciton hopping. a** Top: Schematic of excitons $L$ and $X$, coupled via an effective cavity-mediated hopping of strength $J_{XL}$. Bottom left: exciton energies in the absence of a cavity mode. Bottom center: exciton energies tuned into resonance due to dispersive cavity-dressing of each individual exciton state. Bottom right: Exciton-like system eigenstates as observed in experiment, split due to the effective interaction $J_{XL}$. **b** Left panel: cavity transmission as a function of cavity energy, with the cavity mode positioned at the edge of the domain $L$ as illustrated in the inset. Right panel: derivative of cavity transmission with respect to energy, computed for data in the dashed rectangle in the left panel. The black dashed lines are the eigenstates of the effective system Hamiltonian, Eq. (1). The gray and orange lines are eigenstates of the Hamiltonian in the absence of hopping, $J_{XL} = 0$, corresponding to energies of cavity-dressed but uncoupled $X$ and $L$ excitons, respectively. **c** $J_{XL}$ as a function of the cavity mode position, which is moved away from the etch site center ($x = 0\,\mu$m corresponds to the mode position illustrated in the inset).

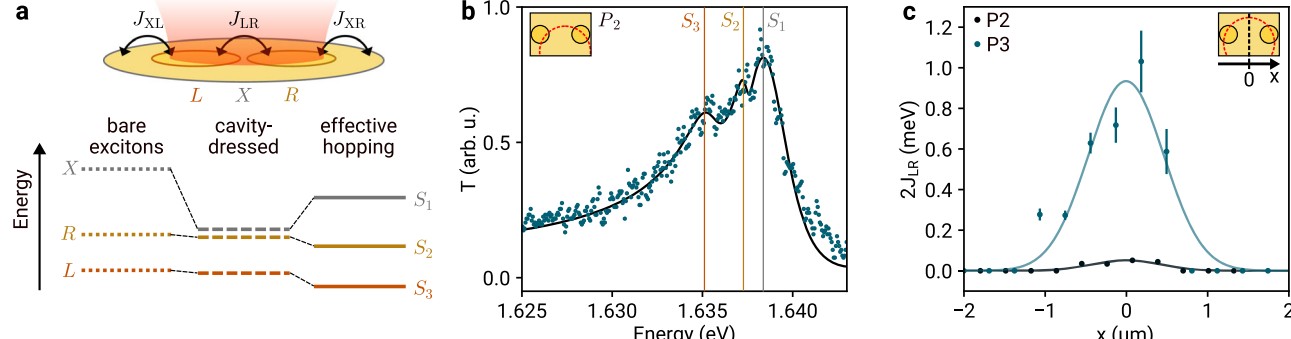

**Fig. 4 | Effective inter-site and site-to-surrounding hopping. a** Coupling $J_{LR}$ between excitons of left and right domain $L$ and $R$, mediated by the cavity in addition to the respective couplings $J_{XL}$ and $J_{XR}$ with the surrounding exciton reservoir $X$. **b** Cavity transmission spectrum (dark green data) for a cavity energy of 1.659 eV and mode position as illustrated in the inset with best fit (black solid line) according to the dissipative model analysis. The vertical lines indicate the eigenenergies of the coupled system. **c** Coupling strength $J_{LR}$ at cavity energy 1.670 eV for the etch site pairs $P_2$ and $P_3$ (black and dark green data, respectively, with error bars of one standard deviation and Gaussian fits as solid lines) as the cavity is moved across the etch site pair, as obtained from exciton light-matter coupling strengths and resonance energies (see Supplementary Note V for details). $x = 0\,\mu$m corresponds to the cavity mode centered between both sites. Error bars indicate one standard error, which for the site $P_2$ is smaller than the respective data points.

(see Supplementary Note V for details). Unavoidably, the coupling vanishes as the cavity is moved away from the etch-site, as confirmed in Fig. 3c.

Finally, we demonstrate effective long-range coupling mediated by the cavity between excitons of two distant sites, as shown conceptually in Fig. 4a: we use cavity-dressing to tune the energy of surrounding excitons $X$ close to the resonances of the disks $L$ and $R$. With dispersive coupling, we obtain hybrid eigenstates $S_1$, $S_2$ and $S_3$ of the coupled system, with energies defined by effective interactions $J_{XL}, J_{XR}$ and $J_{LR}$ among all three constituents. The corresponding spectral signature is shown in Fig. 4b, with maximal intersite coupling strength $2J_{LR}$ of ~0.1 meV and 1 meV obtained for the pairs $P_2$ and $P_3$, respectively. In the framework of dispersive cavity-coupling, the energies of the exciton-like hybrid eigenstates differ from the bare exciton resonances of the system as a direct result of the cavity-induced interaction. This observation is enabled by two features of our system: the energy difference of $L$ and $R$ excitons, a result of our choice of cavity mode position and sample inhomogeneities, renders all three eigenstates optically bright, while the tunability of our fiber cavity provides access to the relevant cavity energy. Lateral displacement of the cavity mode results in the reduction of coupling strength, as expected from Eq. (1) and evidenced by the values for $J_{XL}$ as calculated from exciton energies and light-matter coupling strengths in Fig. 4c.

## Discussion

To conclude, we have developed a deterministic method for dielectric engineering of exciton domains to demonstrate two different features. First, the realization of a cavity-tunable local polariton redshift, which adds a deterministic and flexible strategy to previously demonstrated methods for engineering TMD polaritons via their excitonic fraction[5,18–22]. We anticipate that our approach will enable polariton confinement in two-dimensional cavities to engineer various geometries of polaritonic lattices. As a second key feature, we have demonstrated an effective cavity-mediated long-range exciton hopping in the dispersive coupling regime. This hopping can be implemented in a variety of different exciton systems, such as gate-defined and tunable domains of confined excitons[32,33].

Inhomogeneities in exciton energy and coupling strength in the present device could potentially be mitigated by minimizing local strain, achieved by tip-based smoothing[34] or by filling the through-holes with a different dielectric material to ensure a smooth interface between adjacent layers during transfer[35]. The effective exciton hopping dynamics could be probed and enriched by valley-selective energy transfer[36] in suitable domain geometries and external magnetic fields, enabled by TMD spin-valley locking. Our method could further be applied to different exciton systems, such as Rydberg excitons[37] or hybrid moiré excitons[38,39] with enhanced nonlinearities. Our results thus highlight a promising approach for advancing quantum simulations in lattices and circuits[40] of excitons and exciton-polaritons.

## Methods

### Sample fabrication

Our device is based on a monolayer of $MoSe_2$ grown by chemical vapor deposition and encapsulated in flakes of hBN obtained by mechanical exfoliation. The bottom hBN thickness was 87 nm (as determined with atomic force microscopy), placing the TMD monolayer close to an antinode of the intracavity field. A polymer mask with a through-hole pattern was fabricated with electron beam lithography on the top hBN flake with a thickness of 44 nm. 4% PMMA 950K dissolved in anisole was used as resist, with typical electron beam doses and acceleration voltages of $50\ \mu C/cm^2$ and 20 kV, respectively. Through-holes were etched using inductively coupled plasma-based reactive ion (ICP-RIE) etching. We used a plasma of Ar and $SF_6$, which were injected at flowrates of 5 and 10 sccm, respectively, as the chamber pressure was kept at 10 mTorr. ICP and RF powers were set to 70 W and 6 W, respectively, with a resulting etch rate of 0.6 nm/s. After etching and resist removal, the hBN flake was cleaned using oxygen plasma. Supplementary Fig. 11 shows a scanning electron micrograph of a flake with fabricated through-holes. The complete van der Waals stack was assembled using dry transfer using stamps based on polycaprolactone (PCL) polymer[41,42] to ensure sufficient adhesion to the processed top hBN flake, since transfer attempts based on the widely used polycarbonate were unsuccessful. After stack assembly and release on the cavity mirror, PCL residues were removed using tetrahydrofuran. We note that our fabrication technique also forms the basis for our recent demonstration of a plasmonic metasurface[35].

### Cryogenic cavity system

The fiber-based cavity system, described in detail in ref. 28, was operated in a closed-cycle croystat (attocube attoDRY800) at a base temperature of 4.3 K. The cavity was mounted on a passive vibration isolation system to suppress changes in cavity length induced by mechanical vibrations of the cryostat cold plate. A Gaussian-shaped indentation with radius of curvature of ~14 μm on the fiber tip served as concave cavity mirror. Fiber and planar mirror had identical dielectric coatings, forming highly reflective distributed Bragg reflectors. The cavity transmission was measured with a white-light source (NKT SuperK) in a spectral bandwidth of 10 nm. The cavity mode transmitted through the planar mirror was collimated using an aspheric lens, before being coupled into a single-mode fiber, dispersed in a grating spectrometer and detected with a CCD camera. The cavity also featured higher order Hermite-Gaussian modes[43], whose influence on the transmission measurements is discussed in Supplementary Note II.

The cavity was operated at longitudinal mode order $q = 6$ as the lowest accessible without physical contact between the fiber and the planar mirror (see Supplementary Note VI for details). At this mode order, the measured mode waist was 1 μm, with a 15% in-plane anisotropy. Typical integration times for transmission measurements were chosen between 1 and 2 s in order to average over multiple cryostat compressor cycles. As a result, the measured cavity transmission profiles are broadened by mechanical vibrations. With the cavity mode positioned away from the monolayer, we determined a full-width at half-maximum cavity linewidth $\kappa = 1.52$ meV from the fit of a Voigt profile, with Lorentzian and Gaussian contributions of 1.15 and 0.70 meV stemming from broadening by mirror loss and mechanical vibrations, respectively.

## Data availability

The data that support the findings of this study are available in the main text and the Supplementary Materials. Source data are provided with this paper.

## Code availability

The codes that support the findings of this study are available from the corresponding authors upon request.

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

## Acknowledgements

We thank David Hunger for providing the cavity fiber, as well as Philipp Altpeter and Christian Obermayer for assistance in the clean room. This research was funded by the European Research Council (ERC) under the Grant Agreement No. 772195 as well as the Deutsche Forschungsgemeinschaft (DFG, German Research Foundation) within the Priority Programme SPP 2244 2DMP and the Germany's Excellence Strategy EXC-2111-390814868 (MCQST). I.B. acknowledges support from the Alexander von Humboldt Foundation. L.H. and A.H. acknowledge funding by the Bavarian Hightech Agenda within the EQAP project. F.T.-V. acknowledges funding from the European Union's Framework Programme for Research and Innovation Horizon Europe under the Marie Skłodowska-Curie Actions grant agreement no. 101058981. K.W. and T.T. acknowledge support from the JSPS KAKENHI (Grant Numbers 21H05233 and 23H02052), the CREST (JPMJCR24A5), JST and World Premier International Research Center Initiative (WPI), MEXT, Japan. I.C. acknowledges financial support from the Provincia Autonoma di Trento, from the Q@TN Initiative, and from the National Quantum Science and Technology Institute through the PNRR MUR Project under Grant PE0000023-NQSTI, co-funded by the European Union - NextGeneration EU.

## Author contributions

L.H., F. T-V., and L.K. fabricated samples using monolayer crystals synthesized by I. B. and high-quality hBN crystals provided by K.W. and T.T. L.H. performed the experiments in a cryo-cavity implemented by J.S. and S.V. L.H., I.C., and A.H. analyzed the data. L.H. and A.H. wrote the manuscript. All authors commented on the manuscript.

## Funding

## Competing interests

The authors declare no competing interests.
