## [Transparent Peer Review file · Nature Communications]

Cavity-mediated exciton hopping in a dielectrically engineered polariton system

Corresponding Author: Professor Alexander Högele

Version 0:

Reviewer comments:

Reviewer #1

(Remarks to the Author)

This work reports the modulated interactions between domains of monolayer MoSe₂ excitons via tunable cavity modes. In this tunable cavity system integrated with nano-patterned hBN-encapsulated MoSe₂ van der Waals heterostructures, the interactions between excitonic domains are claimed as long-range exciton hopping in the framework of multiple cavity-coupled-exciton Hamiltonian. This observation is regarded as a premise to quantum simulation architectures, and is prospected to complement the cavity-photon mediated quantum systems of cold atoms, quantum emitters and optomechanics. In comparisons with previous similar works of engineered polariton potential traps that contribute to polariton nonlinearity in Refs. [5, 18-21], this work reports a distinct way of engineering the polariton potential landscapes. For the claim of exciton hopping, it stays in the theoretical representation of the coupling term in the coupled Hamiltonian, but is not really related to any exciton transport/propagation/energy transfer properties. The magnitudes of the potential modulation (up to 1.5 meV) are kind of small in terms of the fluctuation of excitonic inhomogeneous energies (~ 6 meV), making this way of engineering polariton landscapes extremely challenging for scalable quantum architectures. Overall, I cannot recommend the publication of this manuscript at this stage, as I find the claim to be somehow exaggerating and am also concerned with some technical issues. My specific comments are as below.

1. The claim of exciton hopping is based on the coupling term in the coupled Hamiltonian (eq. 1 in the manuscript). This term is the coupling effect of the same cavity mode by different exciton domains, and is mainly characterized by the cavity coupling strengths. The experimental results mainly indicate the cavity coupling could extend to different exciton domains within finite space. However, these results do not include any experimental characteristics of exciton hopping, such as exciton transport, exciton propagation or any energy transfer dynamics. How this coupling term realistically affects or even determine the behaviors of exciton hopping still remains elusive. I find this claim not convincing, or at least over exaggerating.

2. As discussed above, to strengthen the claim of exciton hopping, can the authors perform specific characterizations of the exciton hopping, such as exciton transport, exciton propagation or any energy transfer dynamics experiments?

3. The fluctuation of inhomogeneous exciton potentials largely exceeds the potential modulation depths of the coupling term, making the claim of engineering scalable polariton landscapes less practical. In Fig. 2c, the fluctuation of inhomogeneous exciton potentials is ~ 6 meV, which reflects the realistic potential landscape. However, the inter-exciton-domain coupling term is only maximized ~ 1.5 meV, which is much less than the potential fluctuation and would have limited realistic impact on the potential landscape engineering. Moreover, the potential fluctuations between Fig. 2a and 2b seem to be inconsistent. The SI Fig. 4 also confirms the potential fluctuations of excitons are a major detrimental factor for engineering the polariton potential landscapes.

4. The coupled Hamiltonian is based on a mean-field approximation, where the exciton domains are approximated as individual coupling particles with the same exciton resonances and dissipations. As discussed in Comment 3, the fluctuation of exciton resonances is ~ 6 meV, which is almost at the same order of the coupling strengths of different exciton domains in SI Fig. 4. The authors are strongly suggested to carefully check the validity of this approximation.

5. In comparisons with previous similar works of Refs. [5, 18-21], the coupling seems to be more deterministic in these nano-patterned structures. But the modulation potential depths are much smaller with limited realistic impact on confined polariton interactions. The authors are strongly suggested to carefully compare the impact of this potential engineering with the previous works.

6. For the distant cavity coupling effect, would there be any polariton nonlinear effects? For example, would it be possible to try pump power-dependent measurements to explore the polariton-density dependent coupling strengths and related nonlinear effects?

7. Generally, there are also strong trion resonances below the exciton resonances in monolayer MoSe₂. The exciton resonances ~ 1.640 eV in this work seem to be also close to the reported trion resonances in previous works. Can the authors clarify all these coupling effects are based on the pure excitons? As trion resonances are generally about 20 meV below the exciton resonances, would trions also play a role for the coupling effect with localized excitons (L or R exciton domains)?

8. Raster scans for P1 and P2 in Fig. 2 are along the waist diameter of the cavity mode, but the Raster scans for P2 and P3 in Figs. 3 and 4 do not align with the diameter. Why set the experimental scan schemes differently for these cases? Would the coupling strength of two distant disks be different in these scan schemes?

Reviewer #2

(Remarks to the Author)

Summary: The manuscript by Husel et al. presents a compelling experimental demonstration of engineering exciton-polariton energy landscapes through spatially patterned dielectric environments in monolayer transition metal dichalcogenides (TMDs) coupled to a fiber-based microcavity. The authors successfully show the creation of localized exciton-polariton domains with tunable energies and, crucially, establish cavity-mediated long-range exciton hopping between these distant domains in the dispersive coupling regime. This work represents a significant step forward towards the realization of interacting polaritonic networks using dielectrically tailored 2D semiconductors. The manuscript is well-written, and the findings are of high interest to specialists in exciton-polariton physics, quantum optics, and 2D materials. Overall Recommendation: Although the dielectric engineering of exciton polaritons shown in the first half of the paper is a direct and straightforward consequence of the well-studied dielectric engineering of excitons, the manuscript presents robust and novel results on cavity-mediated exciton hopping over a few microns. The findings are significant and contribute meaningfully to the understanding and control of light-matter strong coupling in 2D materials for future quantum technologies. With revisions as suggested below in the comments, the paper would be suitable for publication in Nature Communications.

Significance: The work builds upon prior studies demonstrating polariton formation in 2D semiconductors and various methods for polariton trapping, such as local disorder, electrostatic tuning, or strain. While the tunable cavity setup utilized here is not compatible with practical applications (e.g. compared to electrostatic tuning), the method is well suited for fundamental research.

The primary result is the demonstration of effective long-range exciton hopping mediated by the cavity in the dispersive coupling regime. This is conceptually illustrated by the Hamiltonian (Eq. 1) which describes effective beam-splitter type coupling between excitons through dispersive exchange of cavity photons. The authors convincingly show both site-to-surrounding coupling (Fig. 3b) and inter-site coupling between distant domains (Fig. 4b), with coupling strengths up to 1 meV. This ability to engineer and observe effective long-range interactions is crucial for building reconfigurable polaritonic circuits

Furthermore, the paper describes a deterministic and tunable control over local exciton-polariton energies achieved by nanopatterning the hBN encapsulation layer, leading to distinct exciton domains. This approach has been previously utilized (10.1103/PhysRevB.108.035427, 10.1038/ncomms15251) to engineer the bandgap of excitons in 2D materials, and is extended here for its application to exciton polaritons. The mesoscopic domains result in an effective attractive polariton potential, tunable with cavity energy, with potential applications for non-linear effects and condensation.

Originality: The originality of this manuscript lies in its approach to deterministically couple dielectrically engineered excitons through cavity coupling by bringing the dressed excitons to resonance using the cavity energy. While other platforms like superconducting qubits and ultracold atoms have demonstrated similar long-range interactions, this work establishes a new pathway within the exciton-polariton platform. This represents a significant advancement by offering a flexible, low-cost fabrication route and leveraging the unique properties of exciton polaritons with large binding energies and interaction strengths.

Comments to main text:

1. Sample Fabrication: Strain and sample imperfections are mentioned in the paper as a possible cause for the changing exciton energies and coupling strengths. Since the sample geometry is critical for the analysis, the authors should show an AFM height map of the sample with the visible etched disks in hBN.
2. Impact of Imperfections: The manuscript acknowledges fabrication imperfections, such as interfacial bubbles, unintentional strain, and spatial energy gradients. While this is acceptable for a fundamental demonstration, the authors could briefly discuss how these imperfections might impact the scalability and uniformity of more complex engineered polaritonic networks in future work, or suggest strategies to mitigate them. This would enhance the practical implications of the work.
3. Strong Coupling and Domain Formation: The evidence for strong coupling is clearly presented through the avoided crossing in cavity transmission, with coupling strengths $g_X = 9.6$ meV for unpatterned regions and $g_L = 2.65 \pm 0.04$ meV for disk-localized excitons. The reduction in coupling for localized domains, consistent with the smaller spatial overlap with the cavity mode, supports the claim of local polariton disk formation. Additionally, I would suggest measuring the reflectance contrast of the heterostructure without the top mirror, to proof that the exciton linewidth is consistent over the whole structure and does not affect the coupling strength in the etched area. It would also allow the authors to correlate the exciton energies at different locations with the exciton energies found by fitting the cavity transmission, and would be a direct proof that your dissipative model for cavity transmission is appropriate for fitting the data.
4. Dielectrically Engineered Energy Modulation: The spatial mapping of polariton energy shifts (Fig. 2a, b, d) clearly demonstrates local energy modulation. The interpretation that the redshift is due to reduced dielectric screening below the hBN holes and corresponds to an attractive polariton potential seems reasonable. However, authors assume that the cavity mode energy is constant over the spatial extent of the scan. However, some areas of the scan contain top hBN and others don't. The large thickness of hBN ($t = 44$ nm) together with its larger refractive index compared to air is enough to shift the cavity mode significantly. In Figure 2a, 2b, the shift in the polariton energies are explained by the coupling of the localized

excitons at lower energies to the cavity mode. The fact that hBN changes the effective cavity length due to its refractive index, however, is not considered when analyzing the shifts in the polariton energies. Could the authors explain why? Is this negligible?

5. Dispersive Regime: The authors address the theoretical requirement for Eq. 1, stating that $|g_i|^2 / \Delta_i^2 \ll 1$ and that deviations from the full Hamiltonian are "well within the typical linewidths observed in the experiments". Could you explicitly state the typical $|g_i|^2 / \Delta_i$ values for the parameters relevant to the hopping demonstrations (Figures 3 and 4)? It seems that the requirements do not always hold but you justify the use of Eq. 1 by comparing its results with numerical simulations. Could you expand more on that and show data that quantifies the agreement between both?

6. Exciton hopping strength: How do you fit the coupling strength vs distance in Figure 3c? Is the theoretical decay proportional to the ratio between the area of the localized and delocalized excitons? How would you optimize the geometry to increase the coupling strength?

Could you show the sweep over cavity length of the transmission, similarly to Figure 3b, from where you extract the data in Figure 4b? These fits are strongly dependent on initial conditions and coefficient bounds. Please provide details on the fit procedure, since the results from Figure 4b,c are the most crucial to the paper. Additionally, please provide error bars for the results on P2 in Figure 4c.

7. Exciton hopping correlation with polariton landscape: Could you give more details on whether/how the attractive polariton potential formed at the etched disk could affect the site-to-surrounding and site-to-site coupling? I don't see a clear connection between the first key feature of the system (tunable redshift of polaritons) and the second feature (cavity dressed exciton coupling).

8. Simulations: The authors should provide evidence that the cavity mode corresponds to the longitudinal mode of order 6 and that the 2D layer is close to an antinode, for instance by showing the simulated electric field distribution for the geometry used.

9. Data Analysis: The use of a dissipative model for cavity transmission (Eq. 2) and the criteria for fitting are well-explained in Supplementary Note II. The discussion of higher-order Hermite-Gaussian modes and their influence on the spectra demonstrates careful consideration of experimental complexities.

Comment so supplementary text:

10. In supplementary note 3, you mention that the spatial variation in the exciton energy is on the order of the typical exciton linewidth. Do you have a PL map showing the spatial distribution of exciton linewidths over the sample, with etched and non-etched areas?

11. In supplementary note 2, you mention that you can deterministically create exciton domains with tunable coupling strength defined by the domain size. This is an interesting way to define areas with different coupling strengths within a single sample. Did you test this hypothesis by measuring coupling strengths vs etch areas of different dimensions?

Minor comments:

12. Top of page 7 "see supplementary note 2 for details" should be "see supplementary note 5 for details".

13. I would suggest adding an x-axis to Figure 1 with cavity mode energies corresponding to cavity lengths. In the text the cavity energy or detuning with respect to the exciton energies is mentioned several times and it helps the reader to visualize that in Figure 1.

14. I would suggest adding the theoretical dispersion of cavity dressed excitons in monolayer and localized areas in Figure 3b for clarity.

15. I would suggest adding the prospect of using the spin-valley physics of TMDCs to probe the inter-site coupling to your conclusions, since the spin-valley lifetime can be similar of larger than the polariton lifetime.

Version 1:

Reviewer comments:

Reviewer #2

(Remarks to the Author)

The authors have addressed all my comments and concerns. I believe the paper is sufficiently rigorous and novel. I therefore recommend the paper for publication in Nature Communications.

Cavity-mediated exciton hopping in a dielectrically engineered polariton system

by Lukas Husel et al.

We thank the reviewers for their time and effort in thoroughly and critically assessing our work. It was very rewarding to see the depth of insight on the side of both reviewers, and we greatly appreciate the critical and constructive feedback of the reviews. Below, we provide a point by point response to the reviewer's comments and suggestions. The corresponding changes are highlighted in the revised manuscript in blue.

Responses to Reviewer #1

This work reports the modulated interactions between domains of monolayer MoSe₂ excitons via tunable cavity modes. In this tunable cavity system integrated with nano-patterned hBN-encapsulated MoSe₂ van der Waals heterostructures, the interactions between excitonic domains are claimed as long-range exciton hopping in the framework of multiple cavity-coupled-exciton Hamiltonian. This observation is regarded as a premise to quantum simulation architectures, and is prospected to complement the cavity-photon mediated quantum systems of cold atoms, quantum emitters and optomechanics. In comparisons with previous similar works of engineered polariton potential traps that contribute to polariton nonlinearity in Refs. [5, 18-21], this work reports a distinct way of engineering the polariton potential landscapes. For the claim of exciton hopping, it stays in the theoretical representation of the coupling term in the coupled Hamiltonian, but is not really related to any exciton transport/propagation/energy transfer properties. The magnitudes of the potential modulation (up to 1.5 meV) are kind of small in terms of the fluctuation of excitonic inhomogeneous energies (~ 6 meV), making this way of engineering polariton landscapes extremely challenging for scalable quantum architectures. Overall, I cannot recommend the publication of this manuscript at this stage, as I find the claim to be somehow exaggerating and am also concerned with some technical issues. My specific comments are as below.

We thank the reviewer for the constructive criticism of our work. Before addressing the specific comments, we briefly summarize the two major observations of our work and our claims regarding their impact and novelty. We hope that this provides a helpful reference in our following responses and serves to avoid misunderstandings.

By nanopatterning the top hBN layer in our device, we realize a deterministic local modulation of the exciton energy. This modulation, with magnitude on the order of 5-10 meV, enables the following key observations:

- I. The deterministic creation of a local lower polariton redshift on the order of 5 -10 meV, as illustrated in Fig. 2. We are convinced that if applied to two-dimensional cavity geometries, this effect will enable the design of polariton potential landscapes as a basis for novel and impactful polariton experiments, even with the level of device inhomogeneities present in the current device.

II. The observation of an effective long-range exciton hopping mediated by our fiber cavity. To the best of our knowledge, this is the first observation of such a hopping for bosonic excitons. As we argue below, the maximum coupling strength between remote disks in the present device would already be sufficient to design an exciton lattice inducing an energy band with a width on the order of the exciton inhomogeneous broadening. Moreover, the strategy can also straightforwardly be applied to design exciton landscapes in a large variety of inorganic and organic quantum well and cavity platforms, where even larger exciton inter-site coupling strengths could be realized by stronger light-matter coupling and smaller mode volumes. As such, our work constitutes a proof of concept experiment, and we are confident that it will inform the design of future cavity-coupled exciton devices.

We agree with the reviewer that when considering such future experiments which build upon our method, a careful analysis of inhomogeneities in our device is crucial. Having performed this analysis in great detail, we conclude that our method can be used to design novel experiments which study the behavior of excitons or polaritons in engineered energy landscapes such as lattices – a scenario frequently referred to as quantum simulation. We are thus convinced that all our above claims are fully sustained, as we argue in detail in the following responses.

1. The claim of exciton hopping is based on the coupling term in the coupled Hamiltonian (eq. 1 in the manuscript). This term is the coupling effect of the same cavity mode by different exciton domains, and is mainly characterized by the cavity coupling strengths. The experimental results mainly indicate the cavity coupling could extend to different exciton domains within finite space. However, these results do not include any experimental characteristics of exciton hopping, such as exciton transport, exciton propagation or any energy transfer dynamics. How this coupling term realistically affects or even determine the behaviors of exciton hopping still remains elusive. I find this claim not convincing, or at least over exaggerating.

and

2. As discussed above, to strengthen the claim of exciton hopping, can the authors perform specific characterizations of the exciton hopping, such as exciton transport, exciton propagation or any energy transfer dynamics experiments?

To elucidate how the effective cavity-mediated exciton coupling term manifests in our system, we consider the scenario of Fig. 3, with two domains of excitons with different mean energies coupled to a single mode of our fiber cavity (see point 4 below for a discussion of the validity of this model in the case of device inhomogeneities). Within this setting, we consider two contrasting cases of spectral detunings, namely the near-resonant and the highly-detuned limits.

The near-resonant case manifests when the cavity is set close to the resonance energies of one or both exciton subsystems. In this case, the total system is described by the light-matter coupling Hamiltonian in Supplementary Eq. 1 of three coupled oscillators (two exciton domains and one cavity photon), with the eigenstates given by three polariton branches with comparable admixtures of the three individual oscillators as quantified by the Hopfield coefficients.

The highly-detuned case is established upon significant detuning of the cavity resonance from the exciton resonances. In this dispersive regime of cavity coupling, the situation changes drastically, and the coupled system is described by Eq. 1 of the main text. In the polariton picture, this regime corresponds to negligible photon Hopfield coefficients. The lower energy states now correspond to two exciton domains, decoupled from the cavity but mutually coupled with strength J via an effective cavity-mediated hopping. This effective coupling is induced by the exchange of virtual cavity photons, and as such is inherently long-ranged: the annihilation of an exciton in one domain results in the creation of an exciton in the other domain (and vice versa) which corresponds to a coherent transfer of excitations between the two exciton domains on the time scale of inverse J .

In our experiments, the implementation of the dispersive cavity-coupling regime is monitored in the spectral domain. In Fig. 3B, the low energy resonances derive from the model of two coupled exciton oscillators, and the cavity photon is entirely absent in this limit. The avoided crossing of strength $2J$, associated with coherent energy transfer between the two exciton oscillators, is spectrally resolved. The observation of this splitting, in conjunction with its quantitative description by the Hamiltonian of the Eq. 1, corresponds to the demonstration of dispersive coupling in the spectral domain.

Complementary experiments of exciton transport or time-domain studies of energy transfer dynamics proposed by the reviewer are challenging yet not necessarily more instructive. The main observable in the time-domain, corresponding to the spectral-domain observables discussed above, would be constituted by oscillations of the exciton populations in the two domains on the maximum time scale of about 430 fs (for $2J = 1.5$ meV). We acknowledge that it could be instructive to study system coherences as well as population transfer among the dispersively coupled exciton subsystems individually on deep sub-micron length scales and fs time scales. However, such experiments are extremely challenging (if feasible at all) and certainly beyond our experimental capabilities.

We conclude our response to the interrelated points 1 and 2 above by noting that the previous studies of cavity-mediated interactions (Refs. [26] and [27]) have been limited exclusively to spectral-domain demonstrations, and wish to emphasize that within a corresponding framework our observations of the key signatures of effective long-range exciton coupling in the dispersive cavity-coupling regime is valid and by no means exaggerated. If the reviewer's concern of exaggeration targets our observation of cavity-mediated interactions, we now state explicitly in the introduction and conclusion of the revised manuscript that the hopping is *effective*. However, if the concern of exaggeration is rather related to the scaling of the inhomogeneous spectral broadening of the subsystems (up to ~ 6 meV) to the coupling strength (up to 1.5 meV) in our experiments, as framed by the following interrelated comments 3 – 5, we refer the reviewer to the changes made to our claims as detailed below.

3. The fluctuation of inhomogeneous exciton potentials largely exceeds the potential modulation depths of the coupling term, making the claim of engineering scalable polariton landscapes less practical. In Fig. 2c, the fluctuation of inhomogeneous exciton potentials is ~ 6 meV, which reflects the realistic potential landscape. However, the inter-exciton-domain coupling term is only maximized ~ 1.5 meV, which is much less than the potential fluctuation

and would have limited realistic impact on the potential landscape engineering. Moreover, the potential fluctuations between Fig. 2a and 2b seem to be inconsistent. The SI Fig. 4 also confirms the potential fluctuations of excitons are a major detrimental factor for engineering the polariton potential landscapes.

We agree with the reviewer that when discussing our method in the context of engineering polariton potentials (point I. in our introductory response above) or arrays of exciton sites with cavity-mediated hopping between them (point II. in our introductory response above), it is crucial to consider device inhomogeneities as well as strategies to mitigate their influence.

First, we discuss the sample inhomogeneity on the basis of confocal hyperspectral maps, as presented in the following figure and now also included as Supplementary Fig. 2.

Response Figure 1 / Supplementary Fig. 2: **a**, Large-area raster-scan map of cryogenic confocal PL intensity, integrated in the spectral range 1.61 to 1.72 eV. The investigated device area, shown in the inset, contains the three etch site areas shown in Fig. 1b of the main text. **b** and **c**, PL FWHM linewidth and energy shift from the mean exciton energy, respectively, for the same sample area as in **a**, obtained by fitting a Lorentzian to the exciton PL spectra. The dashed circles indicate the point spread function of the etch sites.

As evident from Response Fig. 1b, the exciton linewidth away from any processed hBN is ~ 4 meV, indicating near-constant inhomogeneous broadening in the investigated device area. Near the etch sites, the additional redshifted exciton resonance from disk-confined excitons broadens the PL feature. We found that this redshifted resonance has a typical PL linewidth of around 6-10 meV. In Response Fig. 1c, we show the local shift from the mean energy of the exciton PL energy. Local energy shifts are on the order of a few meV, which quantifies the inhomogeneity in the relevant device area.

We now discuss the implications of sample inhomogeneities for the different experimental settings investigated in our work. In Fig. 2, we demonstrate that a local lower polariton redshift

is induced by our fabrication method, which can be used to design polariton potential landscapes. Fig. 2c shows the magnitude of this redshift, which reflects the depth of the respective potential well. When considering an extension of our method to lattices of multiple wells, two sources of disorder are relevant. First, the inhomogeneous exciton broadening on the spatial extent of a single potential well, which defines the polariton linewidth. This linewidth is ~ 6 meV in Fig. 2b for the cavity near-resonant with the excitons, which is the relevant detuning range for polariton experiments. Second, the difference between the redshift induced at the single sites, which is ~ 3 meV for the two different sites in Fig. 2b. Notably, the magnitude of both inhomogeneities are on the order of the local polariton redshift of ~ 6 meV. We therefore expect that a polariton lattice built from our device can be used to modulate the polariton energy within a bandwidth exceeding the sources of disorder. We also point out that similar redshifts of TMD polaritons induced by random dielectric disorder have been sufficient to induce bosonic condensation even in room-temperature broadened polaritons (Ref. [5] of the main text).

Despite these prospects, the inhomogeneities in our device are of course undesired and will ultimately have to be mitigated, which has also been remarked by the reviewer 2. Informed by the feedback of both reviewers, we therefore now discuss strategies to reduce the disorder in our device in the final section of our manuscript. We further point out that when discussing these inhomogeneities in our proof-of-concept experiment, it is also necessary to put our work into context with the existing literature. Previous works on trapping TMD polaritons via their exciton fraction (Refs. [5, 18-22] of the main text, see also our response to comment 5 below) have relied on random disorder potentials or methods with very limited flexibility in trap design. Here, our method constitutes a significant advancement towards a more deterministic implementation of polariton traps in TMDs.

Figs. 3 and 4 of the main text demonstrate signatures of effective cavity-mediated exciton hopping in our device. This mechanism could be exploited to construct lattices of coupled exciton domains. Such an experiment would require more advanced devices to completely suppress the influence of “bare” monolayer excitons (labeled “X” in our manuscript), which could e.g. be achieved by using dielectrics with larger refractive index or nanostructured electrostatic gates. The relevant sources of disorder in this lattice of disk-shaped exciton domains would then be the difference in exciton energy between the sites, around ~ 3 meV in Supplementary Fig. 4, and the exciton inhomogeneous linewidth of around ~ 4 meV. These values are comparable to e.g. the bandwidth $8J$ expected for a rectangular lattice with a hopping strength J on the order of 0.5 meV (maximum value in Fig. 4c) from a tight-binding estimate. We therefore expect that such an energy modulation might already be possible to detect with the present level of disorder. Reducing disorder or increasing the hopping strength via increased light-matter coupling in 2D cavities should enable the realization of exciton energy bands induced by lattices with cavity-mediated hopping in future devices.

4. The coupled Hamiltonian is based on a mean-field approximation, where the exciton domains are approximated as individual coupling particles with the same exciton resonances and dissipations. As discussed in Comment 3, the fluctuation of exciton resonances is ~ 6 meV, which is almost at the same order of the coupling strengths of different exciton domains

in SI Fig. 4. The authors are strongly suggested to carefully check the validity of this approximation.

The validity of the Hamiltonian of Eq. 1 becomes evident by focusing on the non-dissipative, full polariton Hamiltonian of Supplementary Eq. 1:

$$H = E_C a^\dagger a + \sum_i E_i b_i^\dagger b_i + \hbar g_i (a_i^\dagger b_i + b_i^\dagger a_i).$$

This Hamiltonian describes excitonic oscillators in spatially distinct domains in our samples coupled to a single mode of our cavity. It can be derived by Fourier transforming the standard momentum-space representation of the polariton Hamiltonian. Note that in our case, the excitons in the different domains are described as quasi-free particles, since quantum confinement by the change in dielectric environment is negligible. Local fluctuations of the exciton energy and additional sources of inhomogeneous disorder could in principle limit the validity of the Hamiltonian of Supplementary Eq. 1, calling instead for a model describing a reduced number of exciton resonances with effective inhomogeneous broadening. This is clearly not the case in our device, as the different polariton branches described by Supplementary Eq. 1 are resolved in the cavity spectra. The mean field description of Supplementary Eq. 1 is thus adequate to infer the systems resonances, broadened by local disorder.

In the dispersive limit of cavity coupling, Eq. 1 of the main text,

$$H \approx (E_C - \sum_i \frac{g_i}{\Delta_i}) a^\dagger a + \sum_i (E_i + \frac{g_i}{\Delta_i}) b_i^\dagger b_i + \sum_i \sum_{j \neq i} \frac{g_i g_j}{2\Delta_i} (b_i^\dagger b_j + b_j^\dagger b_i),$$

is a valid approximation to the full Hamiltonian and as such is also an adequate description of the system's eigenstates in our device.

5. In comparisons with previous similar works of Refs. [5, 18-21], the coupling seems to be more deterministic in these nano-patterned structures. But the modulation potential depths are much smaller with limited realistic impact on confined polariton interactions. The authors are strongly suggested to carefully compare the impact of this potential engineering with the previous works.

The reviewer is correct that when arguing for novelty and impact of our work, a careful comparison with existing methods is necessary. Related previous work, cited as Refs. [5, 18-22] in our manuscript, reflects the state-of-the-art methods to locally modulate the energy of TMD monolayer polaritons via their excitonic fraction. In our device, the same effect is achieved by a deterministic, local modulation of the exciton energy via engineering of the dielectric environment (see Fig. 2 of the main text and observation I. in our brief summary above).

The first important consideration is the ability of a given method to deterministically modulate the polariton energy. Here, as acknowledged by the reviewer, our method compares favorably with those presented in Refs. [5, 18-22]. The second important consideration, as pointed out by the reviewer, is the magnitude of the local energy shift, which should be sufficient to induce a polariton trapping effect in 2D cavity geometries. In our device, this shift is around 6 meV

when cavity and exciton are near-resonant (Fig. 2c). Refs. [18-22] report larger energy shifts, with maximum values near ~ 40 meV in Ref. [22]. Ref. [5], where trapping is attributed to local disorder of unknown origin, does not report on the trap depth. However, a polariton redshift of 5 meV is observed, which can be attributed to the energy splitting between free and trapped polaritons. Remarkably, this splitting is large enough to induce polariton condensation in the trap at ambient conditions with much larger exciton linewidths than those in our device. We are therefore confident that our method, which generates polariton energy shifts of similar magnitude, can also be used to design polariton traps with sufficient depths to allow for bosonic condensation resulting from interactions of confined polaritons. Based on these considerations, we conclude that compared to Refs. [5, 18-22] our method presents a significant advancement towards the deterministic design of polariton traps with the potential for scalability.

6. For the distant cavity coupling effect, would there be any polariton nonlinear effects? For example, would it be possible to try pump power-dependent measurements to explore the polariton-density dependent coupling strengths and related nonlinear effects?

This is a very insightful remark and indeed a very interesting question how nonlinear behavior of polaritons and excitons could manifest in our system. In the presence of sufficiently strong exciton-exciton interactions, one would expect that coupling of two distant exciton domains via the same cavity mode could indeed constitute nonlinearities in the system in the form of (anharmonic) Josephson oscillations or self-trapping, similar to the observations in GaAs (Abbarchi et al, Nature Physics 9, 275, 2013). In our device, related phenomena could potentially be enabled by density-dependent coupling strengths arising from interaction-induced exciton energy shifts. Estimating the interaction-induced blueshift U using $g_{XX} = 1 \mu\text{eV} \cdot \mu\text{m}^2$ (Emmanuele et al, Nat. Commun 11, 3589, 2020) at exciton densities near 10^{12} cm^{-2} , we find a value of $U=1$ meV, which could allow to reach the nonlinear regime of anharmonic oscillations $U \sim J \geq 1$ meV. Anharmonic Josephson oscillations of this strength could potentially be visualized in pump power-dependent transmission measurements. However, as discussed in our response to comment 1 above, time-resolve pump-probe experiments on fs time scales are beyond our present experimental capabilities.

7. Generally, there are also strong trion resonances below the exciton resonances in monolayer MoSe₂. The exciton resonances ~ 1.640 eV in this work seem to be also close to the reported trion resonances in previous works. Can the authors clarify all these coupling effects are based on the pure excitons? As trion resonances are generally about 20 meV below the exciton resonances, would trions also play a role for the coupling effect with localized excitons (L or R exciton domains)?

The reviewer is correct in noting that cavity-coupled trions in MoSe₂ are known to influence the cavity transmission spectra by forming polaritons. In our device, a small degree of residual doping is indeed present, resulting in the formation of a weak trion resonance red-detuned from the exciton by the trion binding energy of ~ 25 meV. This resonance couples only weakly to the cavity, as evident from cavity resonance sweep in the following figure:

Response Figure 2: Cavity transmission for varying cavity length on an unstructured MoSe₂ monolayer region, with exciton and trion energies labelled as E_X and E_T , respectively.

The large energy detuning between excitons and trions, combined with a small trion light-matter coupling strength, renders the influence of trion-polaritons on the exciton and polariton resonances considered in our work negligible. We have now added this relevant information to Supplementary Note II, stating that “A trion resonance with an energy near 1.62 eV coupled weakly to the cavity is irrelevant for the findings presented in this manuscript due to sizable energy detuning”.

8. Raster scans for P1 and P2 in Fig. 2 are along the waist diameter of the cavity mode, but the Raster scans for P2 and P3 in Figs. 3 and 4 do not align with the diameter. Why set the experimental scan schemes differently for these cases? Would the coupling strength of two distant disks be different in these scan schemes?

The reviewer points to an important detail in our measurement protocol. Indeed, contrary to Fig. 2, the line scans in Figs. 3 and 4 do not cross through the etch sites center. The reason for this is explained in detail in Supplementary Note II. In brief, we found that when scanning the cavity length near the etch site centers, higher transverse order Hermite Gaussian cavity modes contributed additional branches to the spectra, making it hard or even impossible to analyze them with our dissipative transmission model. This effect is visualized in Supplementary Fig. 3, which we reproduce here for the reviewers' convenience:

Response Figure 3 / Supplementary Fig. 3: **a**, Measured cavity transmission for a sweep of the cavity length. The position of the cavity mode with respect to the etch site pair P2 is illustrated in the inset. Excitons in the left etch site and the encapsulated monolayer couple to the cavity, resulting in three polariton branches. **b**, Plot of theoretical cavity transmission (Supplementary Eq. 2), with parameters obtained from a fit to the data in **a**. **c**, Same as in **a**, with a different cavity mode position as illustrated in the inset. Higher order transverse cavity modes contribute additional cavity resonances to the spectra. **d**, Plot of theoretical cavity transmission, with the model of Supplementary Eq. 2 extended to the case of three cavity modes coupling to the excitonic resonances. The model parameters were adjusted to yield agreement with the data in **c**.

In fact, the higher order modes contribute to the middle polariton branches visible in Fig. 2 of the main text, where they are irrelevant to our analysis of the lower polariton energies.

The reviewer is correct in noting that moving the cavity mode closer to the etch site centers should change the light-matter coupling strengths of the exciton oscillators. The coupling strength of the disk-shaped exciton domains is expected to increase due to increased overlap with the cavity mode. This would in turn also increase the strength of the cavity-mediated exciton hopping J .

Responses to Reviewer #2

Summary: The manuscript by Husel et al. presents a compelling experimental demonstration of engineering exciton-polariton energy landscapes through spatially patterned dielectric environments in monolayer transition metal dichalcogenides (TMDs) coupled to a fiber-based microcavity. The authors successfully show the creation of localized exciton-polariton domains with tunable energies and, crucially, establish cavity-mediated long-range exciton hopping between these distant domains in the dispersive coupling regime. This work represents a significant step forward towards the realization of interacting polaritonic networks using dielectrically tailored 2D semiconductors. The manuscript is well-written, and the findings are of high interest to specialists in exciton-polariton physics, quantum optics, and 2D materials.

Overall Recommendation: Although the dielectric engineering of exciton polaritons shown in the first half of the paper is a direct and straightforward consequence of the well-studied dielectric engineering of excitons, the manuscript presents robust and novel results on cavity-mediated exciton hopping over a few microns. The findings are significant and contribute

meaningfully to the understanding and control of light-matter strong coupling in 2D materials for future quantum technologies. With revisions as suggested below in the comments, the paper would be suitable for publication in Nature Communications.

Significance: The work builds upon prior studies demonstrating polariton formation in 2D semiconductors and various methods for polariton trapping, such as local disorder, electrostatic tuning, or strain. While the tunable cavity setup utilized here is not compatible with practical applications (e.g. compared to electrostatic tuning), the method is well suited for fundamental research.

The primary result is the demonstration of effective long-range exciton hopping mediated by the cavity in the dispersive coupling regime. This is conceptually illustrated by the Hamiltonian (Eq. 1) which describes effective beam-splitter type coupling between excitons through dispersive exchange of cavity photons. The authors convincingly show both site-to-surrounding coupling (Fig. 3b) and inter-site coupling between distant domains (Fig. 4b), with coupling strengths up to 1 meV. This ability to engineer and observe effective long-range interactions is crucial for building reconfigurable polaritonic circuits

Furthermore, the paper describes a deterministic and tunable control over local exciton-polariton energies achieved by nanopatterning the hBN encapsulation layer, leading to distinct exciton domains. This approach has been previously utilized (10.1103/PhysRevB.108.035427, 10.1038/ncomms15251) to engineer the bandgap of excitons in 2D materials, and is extended here for its application to exciton polaritons. The mesoscopic domains result in an effective attractive polariton potential, tunable with cavity energy, with potential applications for non-linear effects and condensation.

Originality: The originality of this manuscript lies in its approach to deterministically couple dielectrically engineered excitons through cavity coupling by bringing the dressed excitons to resonance using the cavity energy. While other platforms like superconducting qubits and ultracold atoms have demonstrated similar long-range interactions, this work establishes a new pathway within the exciton-polariton platform. This represents a significant advancement by offering a flexible, low-cost fabrication route and leveraging the unique properties of exciton polaritons with large binding energies and interaction strengths.

We thank the reviewer for the positive and highly constructive feedback on our manuscript. The comments address important aspects of our work, and we believe that the respective changes made to the manuscript and the Supplementary Information have improved the presentation of our findings.

Comments to main text:

1. **Sample Fabrication:** Strain and sample imperfections are mentioned in the paper as a possible cause for the changing exciton energies and coupling strengths. Since the sample geometry is critical for the analysis, the authors should show an AFM height map of the sample with the visible etched disks in hBN.

While we agree with the reviewer that assessing the disorder in our device is crucial when discussing the present limitations, we argue that additional AFM measurements on the present

device would not add quantitative insight. It would mainly allow to visualize local variations in the height of the heterostack, with a surface potentially contaminated after experimental cool-downs, yet not reveal the most relevant parameter of spatial inhomogeneities, namely the inhomogeneities of the exciton resonance energy. We access these inhomogeneities by hyperspectral exciton PL mapping without the top mirror, with data shown in Fig. 2 of the revised Supplementary Information and Response Figure 1 above, which we reproduce in the following as Response Figure 4 for convenience:

Response Figure 4 / Supplementary Fig. 2: **a**, Large-area raster-scan map of cryogenic confocal PL intensity, integrated in the spectral range 1.61 to 1.72 eV. The investigated device area, shown in the inset, contains the three etch site areas shown in Fig. 1b of the main text. **b**, **c** FWHM PL linewidth and energy shift from the mean exciton energy for the same sample area as in **a**, obtained by fitting a Lorentzian to the exciton PL spectra. The dashed circles indicate the point spread function of the etch sites.

Away from etch sites or obvious defects, the PL linewidth in Response Fig. 4b is ~ 4 meV, with only small fluctuations indicating a spatially homogeneous monolayer crystal. At the etch sites, the PL feature broadens due to the addition of a redshifted exciton oscillator, as also evident from Supplementary Fig. 1. The linewidth of ~ 12 meV near the etch sites is consistent with the estimate that the full feature consists of two oscillators with FWHMs of 4 meV and detuned with respect to each other by ~ 5 -10 meV. Additional inhomogeneities related to fabrication imperfections such as unintentional strain near the etch sites might also contribute to local linewidth broadening. Overall, the resonance energy of the exciton PL, with deviation from the mean value shown in Response Fig. 4c, fluctuates by around 5-10 meV across the investigated device area, a typical value for state of the art TMD devices.

2. Impact of Imperfections: The manuscript acknowledges fabrication imperfections, such as interfacial bubbles, unintentional strain, and spatial energy gradients. While this is acceptable

for a fundamental demonstration, the authors could briefly discuss how these imperfections might impact the scalability and uniformity of more complex engineered polaritonic networks in future work, or suggest strategies to mitigate them. This would enhance the practical implications of the work.

We thank the reviewer for this constructive suggestion. In the revised summary of the manuscript, we now state that “inhomogeneities in exciton energy and coupling strength in the present device could potentially be mitigated by minimizing local strain, achieved by tip-based smoothing [34] or by filling the through-holes with a different dielectric material to ensure a smooth interface between adjacent layers during transfer [35].” Here, we refer to two possible strategies which could be attempted to improve device homogeneity: First, “ironing” the device using an AFM tip. Second, filling the air-holes with a different dielectric material, as also implemented in Ref. [40] of the main text. Since a high contrast in refractive index is desirable for sizable confinement, this strategy would profit from encapsulation dielectric with larger refractive index than hBN. Motivated by the reviewer’s suggestion, we now also state more clearly that the effective exciton hopping we observe can in principle “be implemented in a variety of different exciton systems, such as gate-defined and tunable domains of confined excitons [32, 33]”, which allow for local tunability of the exciton sites and thus reduced site-to-site variation in the exciton energy.

3. Strong Coupling and Domain Formation: The evidence for strong coupling is clearly presented through the avoided crossing in cavity transmission, with coupling strengths $g_X = 9.6$ meV for unpatterned regions and $g_L = 2.65 \pm 0.04$ meV for disk-localized excitons. The reduction in coupling for localized domains, consistent with the smaller spatial overlap with the cavity mode, supports the claim of local polariton disk formation. Additionally, I would suggest measuring the reflectance contrast of the heterostructure without the top mirror, to proof that the exciton linewidth is consistent over the whole structure and does not affect the coupling strength in the etched area. It would also allow the authors to correlate the exciton energies at different locations with the exciton energies found by fitting the cavity transmission, and would be a direct proof that your dissipative model for cavity transmission is appropriate for fitting the data.

As suggested by the reviewer, we have probed the optical response of our device without the second cavity mirror in confocal DR, with the main results presented in the following Response Figure 5:

Response Figure 4: **a**, Differential reflectivity (DR), measured in confocal cryogenic spectroscopy along the etch site pair P2. The gray, yellow and orange dots show the energies of X, L and R excitons, respectively, as obtained from cavity transmission spectra. **b**, DR spectrum with the optical spot placed on the left hole of the etch site pair P2. **c**, Same as **a** but for the etch site pair P3. **d**, Same as **b** but for the right hole of the etch site pair P3.

The main findings of our analysis are as follows: First, we observe similar linewidths for the resonances corresponding to the etch site and the surrounding exciton, indicative of similar inhomogeneous broadening. Second, the difference in the contrast is consistent with the smaller area of the effective exciton disk as compared to its surrounding. As anticipated by the reviewer, this allows us to conclude that local nanopatterning indeed does not affect the coupling strength if scaled by the effective exciton area. Moreover, both the DR and PL exciton energies correspond to the resonance energies derived from the analysis of the cavity transmission. We show this correspondence explicitly in the new Supplementary Fig. 6, reproduced below, where the exciton PL energy is plotted together with the energies extracted from the analysis of the cavity transmission. The blueshift of excitons in the encapsulated region is clearly reproduced, and the spectral position of a redshifted shoulder matches with results of the energies inferred for the exciton domains. Overall, we conclude that the dissipative model we employ is appropriate to infer the excitonic response from measurements of the cavity transmission.

Response Figure 5 / Supplementary Fig. 6: **a**, Confocal PL linecut for P2, measured along the dashed line in Supplementary Fig. 1a, along with exciton resonance energies determined from the analysis of the cavity transmission spectra as shown in Supplementary Fig. 4a (colored dots, position axis scaled and shifted to match the PL emission profile). **b**, Same as **a** but for the etch site pair P3.

4. Dielectrically Engineered Energy Modulation: The spatial mapping of polariton energy shifts (Fig. 2a, b, d) clearly demonstrates local energy modulation. The interpretation that the redshift is due to reduced dielectric screening below the hBN holes and corresponds to an attractive polariton potential seems reasonable. However, authors assume that the cavity mode energy is constant over the spatial extent of the scan. However, some areas of the scan contain top

hBN and others don't. The large thickness of hBN ($t = 44\text{nm}$) together with its larger refractive index compared to air is enough to shift the cavity mode significantly. In Figure 2a, 2b, the shift in the polariton energies are explained by the coupling of the localized excitons at lower energies to the cavity mode. The fact that hBN changes the effective cavity length due to its refractive index, however, is not considered when analyzing the shifts in the polariton energies. Could the authors explain why? Is this negligible?

The shift of the cavity energy induced by the air-holes is estimated to be negligible: the hole radius is $0.3\ \mu\text{m}$, smaller than the waist of $\sim 1\ \mu\text{m}$, such that the hole covers only roughly 10% of the full mode area. When considering its effect on the cavity mode, it should therefore be pictured as a Rayleigh/Mie scattering center. Correspondingly, we clearly observe a reduction in cavity transmission and an increase in cavity linewidth at the position of the etch sites. While scattering centers induce a dispersive shift to the cavity mode in principle, it is expected to be smaller than the polariton linewidth in the present device. For comparison, Ref. 42 of the main text finds that $\sim 50\ \text{nm}$ diameter gold nanoparticles, whose refractive index contrast with air is much larger than that of hBN, induce a shift to the cavity mode on the order of 1 GHz or $3\ \mu\text{eV}$. For air-holes, with increased diameter yet much smaller contrast in the refractive index as compared to Au, we anticipate that the respective effects of the etch sites on the cavity energy should be negligible.

5. Dispersive Regime: The authors address the theoretical requirement for Eq. 1, stating that $|g_i|^2 / \Delta_i^2 \ll 1$ and that deviations from the full Hamiltonian are "well within the typical linewidths observed in the experiments". Could you explicitly state the typical $|g_i|^2 / \Delta_i^2$ values for the parameters relevant to the hopping demonstrations (Figures 3 and 4)? It seems that the requirements do not always hold but you justify the use of Eq. 1 by comparing its results with numerical simulations. Could you expand more on that and show data that quantifies the agreement between both?

The validity of the approximation Eq. 1 is an important aspect of our work, and we now address it in more detail in the revised Supplementary Note V. In particular, we have added a new Supplementary Fig. 8, reproduced here as Response Figure 7:

Response Figure 6 / Supplementary Fig. 8: **a**, Eigenstates of the effective Hamiltonian for the dispersive regime of cavity coupling (Eq. 1 of the main text, solid lines) and of the full polariton Hamiltonian of Supplementary Eq. 1 (dashed lines). The parameters used for the computation correspond to the experimental setting in Fig. 3 of the main text and are given in Supplementary Note V. In the framework of dispersive cavity coupling, the gray (orange) branch is predominantly X-(L)-exciton-like. **b**, g_i^2 / Δ_i^2 , computed for the two exciton domains X and L. The dashed vertical line in both panels indicates the cavity energy at which the experimental branches exhibited minimum splitting J_{XL} .

As we now state in the revised Supplementary Note V, panel a in this figure "shows the eigenstates of both the full polariton Hamiltonian of Supplementary Eq. 1 (dashed lines) as

well as for the dispersive Hamiltonian Eq. 1 of the main text (solid lines). The computation was performed using the parameters for the theoretical prediction shown in Fig. 3b. The agreement between the two models underlines the validity of the approximation even at cavity lengths where $|g_i / \Delta_i|^2 \ll 1$ does not strictly hold (with $|g_i / \Delta_i|^2 < 0.1$ in Supplementary Fig. 8b).”

6. Exciton hopping strength: How do you fit the coupling strength vs distance in Figure 3c? Is the theoretical decay proportional to the ratio between the area of the localized and delocalized excitons? How would you optimize the geometry to increase the coupling strength?

As we discuss in the revised Supplementary Note V, “to obtain the values for the cavity-mediated exciton hopping J_{XL} shown in Fig. 3c of the main text from sweeps of the cavity length as shown in Fig. 3b, we identified zeros in the derivative of the cavity transmission as resonance energies. We then fitted the energy $E_{S1/S2}$ of the two branches with a model for two coupled oscillators, $E_{S1/S2} = (E_{X1} + E_{X2})/2 \pm \text{Sqrt}(J_{XL}^2 + (E_{X1} - E_{X2})^2/2)$, with E_{X1} a constant and E_{X2} a linear function of the cavity length, to obtain J_{XL} .”

The hopping strength J is proportional to the light-matter coupling strengths g_i in the two exciton domains which have overlap η_i with the cavity mode, such that the theoretical decay is proportional to $\text{Sqrt}(\eta_1 \cdot \eta_2)$. To optimize the coupling strength, a minimal distance between neighboring domains should therefore be chosen in order to ensure maximum overlap of each domain with the cavity mode. The domain size should further be maximized to obtain maximum values for the values of g_i , which could be further increased by minimized mode volumes in two-dimensional cavity-geometries. Also, differences in exciton energy between the domains should be minimized, potentially achieved via the strategies discussed in our response to reviewer comment #2 above.

Could you show the sweep over cavity length of the transmission, similarly to Figure 3b, from where you extract the data in Figure 4b? These fits are strongly dependent on initial conditions and coefficient bounds. Please provide details on the fit procedure, since the results from Figure 4b,c are the most crucial to the paper. Additionally, please provide error bars for the results on P2 in Figure 4c.

In the newly added Supplementary Fig. 9, shown as Response Figure 8 below, we now show a cavity length sweep for the experimental configuration of Fig. 4b. In the revised Supplementary Note V, we now state that “in Supplementary Fig. 9a, we show a cavity length sweep obtained at the same position. The dashed black lines show the evolution of the branches computed according to Eq. 1, with the exciton parameters [...] close to the values obtained from the dissipative model fit at the respective position. For cavity energies above 1.651 eV, the model of Eq. 1 is in good agreement with the experimental observations, placing our system in the dispersive cavity-coupling regime. Operation in this regime is confirmed by small values of $|g_i / \Delta_i|^2$ shown in Supplementary Fig. 9b, similar to Supplementary Fig. 8b discussed above.”

Response Figure 7 / Supplementary Fig. 9: Dispersive cavity regime for two coupled exciton disks. **a**, Left panel: cavity transmission as a function of the cavity energy, with the cavity mode covering domains L and R as illustrated in the inset. Right panel: derivative of cavity transmission with respect to energy, computed for the data in the dashed rectangle in the left panel. The dashed lines are the eigenstates of the effective system Hamiltonian, Eq. 1 of the main text, computed for the parameters given in Supplementary Note V. The data are taken at the same position as the spectrum in Fig. 4b of the main text. **b**, g_i^2/Δ_i^2 , computed for the three exciton resonances X, L and R visible in **a**. The black dashed line indicates the cavity energy at which the values for J_{XL} in Fig. 4c of the main text were obtained. The gray dashed line indicates the cavity energy for the transmission spectrum shown in Fig. 4b of the main text.

The values of J_{XL} in Fig. 4c were calculated from the values for E_i and g_i obtained from fits to the cavity transmission spectra, which we had stated in the respective figure caption and described in detail in Supplementary Note V. The resulting error bars for the site pair P_2 in Fig. 4c were smaller than the size of the data points, which we now also state explicitly in the figure caption. For clarity, we now refer to Supplementary Note V in the caption of Fig. 4c and have also added to the main text the statement that this figure shows “... J_{XL} as calculated from exciton energies and light-matter coupling strengths...”

7. Exciton hopping correlation with polariton landscape: Could you give more details on whether/how the attractive polariton potential formed at the etched disk could affect the site-to-surrounding and site-to-site coupling? I don't see a clear connection between the first key feature of the system (tunable redshift of polaritons) and the second feature (cavity dressed exciton coupling).

In the present device implementation, both key features, namely the tunable redshift of polaritons and the cavity dressed exciton coupling, arise from a local, dielectrically induced redshift ΔE_X of local exciton energies in the disks from the surrounding excitons. Deeper attractive polariton potentials would result in larger local polariton redshifts, thus separating spectrally local excitons from their surrounding. This ensures for both site-to-surrounding and site-to-site coupling, that distinct entities of excitons are locally and spectrally defined. Both key features are thus connected in the sense that larger polariton redshift implies spectrally resolved exciton hopping and vice versa.

Motivated by the reviewer's comment, we emphasize now in the final paragraph the two different key features of our system by stating: “... we have developed a deterministic method for dielectric engineering of exciton domains to demonstrate two different features of our system. First, the realization of a cavity-tunable local polariton redshift, which adds a deterministic and flexible strategy to previously demonstrated methods for engineering TMD polaritons via their excitonic fraction [5, 18–22]. We anticipate that our approach will enable polariton confinement in two-dimensional cavities to engineer various geometries of polaritonic lattices. As a second key feature, we have demonstrated an effective cavity-mediated long-range exciton hopping in the dispersive coupling regime.”

8. Simulations: The authors should provide evidence that the cavity mode corresponds to the longitudinal mode of order 6 and that the 2D layer is close to an antinode, for instance by showing the simulated electric field distribution for the geometry used.

We have added Supplementary Note VI, which includes the following newly added Supplementary Fig. 10:

Response Figure 8 / Supplementary Fig. 10: **a**, Experimental transmission spectrum of the empty fiber cavity. The effective cavity length $L_{\text{eff}} = 4.308 \mu\text{m}$ was determined as described in the main text. The two peaks correspond to the resonances of the fundamental transverse modes with different effective longitudinal mode order q . **b**, Same as **a** but for $L_{\text{eff}} = 3.932 \mu\text{m}$. **c**, Intensity of the intracavity light field along the optical axis from a transfer matrix calculation (solid red line). The assumed mirror distance is $1.600 \mu\text{m}$, yielding a resonance wavelength of 764 nm . The refractive index profile along the structure is shown in blue. The vertical dashed line indicates the position of the TMD monolayer, encapsulated between two hBN layers.

Based on this figure, we now describe how we infer the longitudinal mode order from cavity transmission spectra and discuss the simulated electric field distribution along the optical axis of our cavity structure.

9. Data Analysis: The use of a dissipative model for cavity transmission (Eq. 2) and the criteria for fitting are well-explained in Supplementary Note II. The discussion of higher-order Hermite-Gaussian modes and their influence on the spectra demonstrates careful consideration of experimental complexities.

We thank the reviewer for this acknowledgement.

Comment so supplementary text:

10. In supplementary note 3, you mention that the spatial variation in the exciton energy is on the order of the typical exciton linewidth. Do you have a PL map showing the spatial distribution of exciton linewidths over the sample, with etched and non-etched areas?

With reference to our response to the comment #1 above, we included the hyperspectral data from PL mapping in Supplementary Note I, where we also added the statement: “in

Supplementary Fig. 2b, we show the PL FWHM linewidth for the same device area, obtained by fitting a Lorentzian to the exciton PL spectra. At the etch sites, the spectra exhibit spectral broadening, originating from the additional resonance of the redshifted domain-localized excitons. Potential additional inhomogeneous broadening could stem from local disorder induced during the fabrication process. Away from the etch sites, the PL linewidth is homogeneous across the investigated area.”

11. In supplementary note 2, you mention that you can deterministically create exciton domains with tunable coupling strength defined by the domain size. This is an interesting way to define areas with different coupling strengths within a single sample. Did you test this hypothesis by measuring coupling strengths vs etch areas of different dimensions?

We agree with the reviewer that this is a highly interesting perspective to locally tailor the light-matter coupling, which we have not explored so far.

Minor comments:

12. Top of page 7 “see supplementary note 2 for details” should be “see supplementary note 5 for details”.

We thank the reviewer for pointing out this erroneous statement, which we have now corrected.

13. I would suggest adding an x-axis to Figure 1 with cavity mode energies corresponding to cavity lengths. In the text the cavity energy or detuning with respect to the exciton energies is mentioned several times and it helps the reader to visualize that in Figure 1.

We thank the reviewer for this useful suggestion, which we have implemented for improved clarity of the presentation. The new version of the figure is as follows:

Response Figure 9: New version of Fig. 1 of the main text.

14. I would suggest adding the theoretical dispersion of cavity dressed excitons in monolayer and localized areas in Figure 3b for clarity.

We thank the reviewer for this suggestion, which we have implemented to improve the clarity of the presentation in the revised manuscript.

Response Figure 10: New version of Fig. 3 of the main text.

15. I would suggest adding the prospect of using the spin-valley physics of TMDCs to probe the inter-site coupling to your conclusions, since the spin-valley lifetime can be similar of larger than the polariton lifetime.

We thank the reviewer for this highly insightful suggestion. In the final paragraph of the revised manuscript, we now state: “The effective exciton hopping dynamics could be probed and enriched by valley-selective energy transfer [36] in suitable domain geometries and external magnetic fields, enabled by TMD spin-valley locking”, referring to our work on valley-selective energy transfer as reported in the newly included Ref. [36].

List of changes

All changes made to the manuscript and the Supplementary Information are highlighted in the revised versions in blue.

Additional changes

1. We have corrected the cavity mode positions in the insets of Fig. 1e and Fig. 4b.
2. We have corrected the x-axis position of the PL resonance energies indicated by the white dots in Supplementary Fig. 1b, which had erroneously been shifted by one position unit/pixel.
3. In Figs. 3a and 4a, we have replaced the label for “cavity-coupled” excitons by “effective hopping” to clearly emphasize this regime of dispersive cavity-coupling.
4. We have corrected a typo in the final sentence of the introduction.